# System-Level Assessment of Ka-Band Digital Beamforming Receivers and Transmitters Implementing Large Thinned Antenna Array for Low Earth Orbit Satellite Communications

**DOI:** 10.3390/s25154645

**Published:** 2025-07-26

**Authors:** Giovanni Lasagni, Alessandro Calcaterra, Monica Righini, Giovanni Gasparro, Stefano Maddio, Vincenzo Pascale, Alessandro Cidronali, Stefano Selleri

**Affiliations:** 1Department of Information Engineering, University of Florence, Via di Santa Marta, 3, 50139 Florence, Italy; giovanni.lasagni@unifi.it (G.L.); monica.righini@unifi.it (M.R.); stefano.maddio@unifi.it (S.M.); stefano.selleri@unifi.it (S.S.); 2Thales Alenia Space—Italy, Via Saccomuro 24, 00131 Rome, Italy; giovanni.gasparro@thalesaleniaspace.com (G.G.); vincenzo.pascale@thalesaleniaspace.com (V.P.)

**Keywords:** Low Earth Orbit (LEO) satellite communications, non-terrestrial networks, Internet of Things (IoT), Ka-band

## Abstract

In this paper, we present a system-level model of a digital multibeam antenna designed for Low Earth Orbit satellite communications operating in the Ka-band. We initially develop a suitable array topology, which is based on a thinned lattice, then adopt it as the foundation for evaluating its performance within a digital beamforming architecture. This architecture is implemented in a system-level simulator to evaluate the performance of the transmitter and receiver chains. This study advances the analysis of the digital antennas by incorporating both the RF front-end and digital sections non-idealities into a digital-twin framework. This approach enhances the designer’s ability to optimize the system with a holistic approach and provides insights into how various impairments affect the transmitter and receiver performance, identifying the subsystems’ parameter limits. To achieve this, we analyze several subsystems’ parameters and impairments, assessing their effects on both the antenna radiation and quality of the transmitted and received signals in a real applicative context. The results of this study reveal the sensitivity of the system to the impairments and suggest strategies to trade them off, emphasizing the importance of selecting appropriate subsystem features to optimize overall system performance.

## 1. Introduction

Non-terrestrial networks (NTNs) represent a transformative approach in today’s communication systems, utilizing satellite constellations, high-altitude platforms (HAPs) and unmanned aerial vehicles (UAVs) or drones. When integrated with terrestrial networks, these technologies constitute the next generation of communication environments [1].

The advancement of NTNs has been driven by an increasing demand for global connectivity, low-latency internet, and secure communications in remote or hostile environments. Within the realm of satellite communications, this paradigm shift has notably highlighted the utilization of Low Earth Orbit (LEO) satellites. Compared to traditional Geostationary Orbit (GEO) systems, LEO satellites provide several significant benefits, including lower propagation delays, reduced signal losses, and access to wider bandwidth. The lower altitudes of LEO satellites not only contribute to better performance but also result in cost savings in each satellite launch. This evolution has fostered a trend towards the establishment of large-scale constellations composed of low-cost satellites utilizing Commercial Off-The-Shelf (COTS) components, since the impact of a single satellite failure becomes less critical.

Table 1 presents a summary of the constellation deployed for the Internet of Things (IoT), together with the corresponding operator and the number of satellites.

From a technical perspective, the coverage requirements of LEO systems, due to their lower orbital altitude, call for a high number of beams operating at very wide scanning angles to effectively expand the coverage area. In this context, direct radiating arrays are preferred due to their ability to perform electronic scanning, which improves the flexibility and responsiveness of the system in managing coverage. The radiating properties of the array in terms of grating lobes and side-lobe level (SLL), as well as the beamforming architecture adopted, are also critical. Nowadays, there are mainly three architectures for beamforming: analog, digital, and hybrid. In the literature, there are many comparisons between these approaches [2,3,4], each with their benefits and drawbacks:Analog beamforming makes use of dedicated ICs to generate each beam by means of variable attenuators and phase shifters. The main drawback of this architecture is the hardware proliferation and complexity scaling with the number of beams, limiting them to a few tens.Full-digital beamforming feeds each radiating element by up-converting a phase- and amplitude-modulated base-band signal processed in a digital domain. This way it is possible to improve the number of beams with arbitrary directions, with limitations due to processed bandwidth and power consumption.Hybrid architecture is a combination of analog and digital beamforming: the array is viewed as composed of digitally driven sub-arrays in each of which analog beamforming is implemented. The main drawback is due to the limited instantaneous field-of-view achievable.

While analog beamforming is the most consolidated and assessed [5,6], the request for a growing number of beams for these applications is shifting the attention to digital beamforming. To the authors’ knowledge, while some preliminary studies have been conducted [7,8], an investigation into a practical design on the effect of impairments on the performance of a digital antenna is not present in the literature.

In this paper, we firstly consider typical performances of active antennas for LEO telecommunications (see Table 2, [9]) in Ka-band to propose a suitable direct radiating antenna. The satellite size imposes demanding antenna requirements in terms of size, weight, power, and cost; hence, a minimization of the number of elements is desirable, while maintaining high scanning angles and a high side-lobe level, leading to a challenging design. Then, we perform an investigation of a digital chain relying on typical performances of commercially available components to assess how the impairments and non-idealities of such components affect the pattern and link performances, in a framework of a digital-twin model.

The paper is organized as follows: Section 2 presents the study of the irregular active lattice architecture, the optimization strategy adopted and the inter-element analysis of mutual coupling; Section 3 presents the system-level architecture of the whole device, analyzing the quality of the signal thanks to some key performance parameters: Error Vector Magnitude (EVM) and bit error rate (BER) for transmitting and receiving, respectively. Eventually, Section 4 draws conclusions and highlights future work.

## 2. Antenna Synthesis and Modeling

In this section we discuss the study, the analysis, and the design of a dual circular polarized antenna array working in Ka-band, suitable for receiving and transmitting LEO satellite communications. The optimal lattice and feed tapering are reported, and an insight into the mutual coupling between array elements is provided.

### 2.1. Optimum Lattice Architecture

The antenna lattice architectures must be devised as the best trade-off between performances and reduced number of elements, this latter constraint being motivated by the need to limit the number of active channels in the receiver and transmitter architectures.

Whilst a regular circular lattice would have good scanning capabilities, the number of elements is high and the SLL is poor unless amplitude tapering is exploited. Amplitude tapering is generally undesired since, especially if there are substantial differences in the amplitudes, it reduces both aperture efficiency and power amplifier (PA) efficiency.

Possible solutions for uneven yet semi-regular dispositions of elements like the non-uniform sunflower [10] would allow for SLL control without tapering, yet with somewhat inferior scanning capabilities and still a non-negligible amount of elements.

The proposed approach is hence that of a thinned array [11,12,13,14,15,16,17]. In this kind of array, the optimum lattice is found by removing specific elements from the regular lattice. Such a removal, if optimized, leads to an intrinsic amplitude tapering (elements are generally more scattered at the outskirts of the array) and, if amplitude tapering proves necessary for SLL control anyway, the dynamic range, the ratio between maximum and minimum feed amplitude, proves to be generally much lower than for the full array, thanks to sparsity, and hence much less critical for PA efficiency.

Furthermore, thinning is robust with respect to scanning since the thinned lattice is somewhat irregular and this is known to provide good scanning [17,18,19,20,21]. It is, anyway, important to notice that, backing the antenna, a beamforming network (BFN) section will be needed, and some regularity in the element deployment would be preferable to reduce costs, industrialization, testability, and installability [22,23,24]. This is again at a premium on a thinned array as compared to a random element distribution. Furthermore, in this paper, a modular approach has been implemented, by forcing the optimized solution to comprise a central disk and 12 identical sectors [25,26,27].

The basic concept is to start from a regular array with elements deployed in concentric rings. Given the desired distance between elements $d_{0}$, a central element is surrounded by other elements deployed on circles of radii $nd_{0}$ with n=1,…,M. In the present case $d_{0}=0.5\lambda_{0}$ is considered, $\lambda_{0}$ being the free space wavelength at central frequency. M=13 is necessary to comply with beam-width requirements, leading to a $N_{E}=566$ radiating elements array (Figure 1a).

The resulting antenna is then divided into 13 sectors: a circular central sector $S_{0}$, with 7 elements, and 12 identical peripheral sectors, named $S_{I}$ to $S_{XII}$, each comprising 42 elements. The chosen number of sectors, 13, is a trade-off between a minimum number of different sectors and a minimum number of elements for each sector. This choice also imposes that in each ring the number of elements must be a multiple of 12; therefore, some elements are removed and rearranged so that no elements cross, or are too close to a sector border, leading to a starting point for thinning with $N_{E}=511$ elements (Figure 1b).

The sectored array is then the starting point for a multi-objective (MO) optimization based on the binary genetic algorithm (BGA) [28,29,30]. BGA is, of course, the best choice for an optimization problem where the variable space domain exhibits just two states (on and off) [29]. On the other hand, MO allows us to obtain a Pareto set—an approximation of the problem’s Pareto front—containing the non-dominated solutions with respect to a set of conflicting requirements. In our case, the requirements were the lowest number of elements and the best SLL over the entire scan range.

Several MO runs were carried out and lattices were evaluated by analyzing Pareto sets. The chosen case for further investigations (Figure 1a) exhibits $N_{E}=319$, that is a substantial 44% reduction in the number of elements, and a 20 dB SLL, 7 dB better with respect to the full array of Figure 1a. While the number of elements is fine, SLL could still be improved; hence, an amplitude tapering is considered. This is implemented as a function of the distance from the center. The number of rings, starting from the outer one, with variable amplitude values, is a design parameter.

In this case, due to the continuous nature of the amplitude tapering, an Invasive Weed Optimization (IWO) is exploited [30,31,32,33] in a single objective framework, optimizing only for the SLL. Fixed amplitudes are considered unitary; variable amplitudes of outer rings span the range [0, 2], to also allow for elements with higher feed amplitudes, if necessary. A parametric study is carried out with the number of outer rings $N_{r}$ whose amplitude feed is to be optimized in the set 5,8,10,12.

Table 3 summarizes the results. The number of concentric rings, from outer to inner, is given in the first column; the second column reports the number of elements with a tapered amplitude; the third column shows the attained SLL; the fourth column shows the dynamic range of tapering, the ratio between maximum and minimum amplitude.

The best configuration is the one with 12 rings since it attains almost 2 dB better SLL and exhibits a limited amplitude dynamic, slightly above 3. Full results are shown in Table 4 and Figure 2 and Figure 3. The array geometry is shown with feed amplitudes in a color scale from lower (blue) to higher (red). The 3D broadside pattern and the two principal cut planes are also reported. In addition SLL attained in scanning is reported in Figure 4. Scanning has been done for an extended θ range (up to $57^{\circ }$) range and in one-eighth of the azimuthal range, spanning more than a single sector and is hence representative of the complete azimuthal range.

Since tolerance might have an adverse effect on SLL performances, a series of Monte Carlo statistical analyses were performed, considering the radiators’ position tolerances, due to the etching process, up to 15 µm and considering the assembling tolerances of the sectors up to 150 µm. The average SLL obtained was −21.71dB while the degraded case in 200 independent analyses was −21.56dB, therefore less than 0.3, dB worse than the nominal case.

### 2.2. Radiating Element Analysis

The radiating element selected for the array implementation is depicted in Figure 5. It is a multilayer, dual-polarized, slot-coupling-fed radiator comprising two stacked circular patches in order to span the required 11.9% bandwidth at 29.25 GHz. The stack-up, top to bottom, comprises a larger 2.79 mm diameter circular patch, etched on the bottom of a 0.127 mm thick Rogers RT-5880 ($\epsilon_{r}=2.2$, tanδ=0.0009) substrate [34]; a 0.508 mm thick Rogers RT-5880 layer; a smaller 2.045 mm diameter circular patch, etched on the top of a 0.254 mm thick Rogers RO4350B ($\epsilon_{r}=3.66$, tanδ=0.0037) substrate (Figure 5a) [35]; a ground plane with a pair of C-shaped feeding slots; a 0.245 mm thick Rogers RT6006 ($\epsilon_{r}=6.15$, tanδ=0.0027) substrate [36]; the feeding micro-strip which, finally, leads to coaxial connectors (Figure 5a). The two feed points guarantee polarization agility. The full description of the element is out of the scope of this paper. The reader can refer to [37] for the design of multilayer, slotted feed radiating elements. Element performances, in terms of matching and patterns, are shown in Figure 5c,d. For brevity, results concern only one of the two feeding ports. A slight asymmetry in the pattern on one of the two planes is due to the asymmetry in the slot position beneath the patches.

Furthermore, Monte Carlo simulations were performed to quantify the sensitivity of the antenna’s radiation characteristics to manufacturing tolerances, adopting the same statistical framework used in the SLL study. Realistic variability ranges for both material properties and fabrication processes were applied. The resulting reflection-coefficient statistics appear as a family of $|S_{11}|$ curves; Figure 5c shows the mean response and the ±1σ envelopes. An analogous analysis for the radiation pattern produces a set of gain profiles, but, in this case, the dispersion is negligible. Indeed, all patterns differ by at most a fraction of a decibel, hence only the mean curve for the two principal radiation cuts is reported in Figure 5d.

#### Inter-Element Coupling

To assess the mutual coupling between array elements, full-wave electromagnetic simulations were performed in the time domain using the methodology described in [38], with each element excited via a coaxial port. The analysis was restricted to selected portions of the array—specifically, the central core for the $S_{0}$ elements (Figure 6a) and a configuration including three adjacent sectors plus the central one (Figure 6b). This simplification is based on the assumption that coupling with non-adjacent sectors is negligible compared to that with immediate neighbors. As a result, the simulation domain is significantly reduced, yielding savings in both computational time and memory usage. The objective of this analysis is to extract an accurate electromagnetic model of the radiating element—capturing both impedance matching and radiation characteristics—in the presence of neighboring elements and edge effects. This model is subsequently used to feed the commercial CAD tool employed in the system-level performance evaluation presented in the next section.

Figure 7 presents a selection of the results for mutual coupling: Figure 7a shows mutual coupling in $S_{0}$ when port 1 is fed; Figure 7b shows mutual coupling when port 15 is fed; this analysis encompasses both setups in Figure 6. Even if only a very limited set of curves is reported for the sake of brevity, these are representative of the global behavior and mutual coupling never exceeds −14 dB over the whole bandwidth.

The result of this analysis provides a coupling matrix which will be exploited in the system-level simulator to account for the effects in matching and radiation characteristics of the single elements, non-isolated, when evaluating the performance of the array, also including alterations of the radiation pattern and SLL degradation.

## 3. Model of the Digital Beamforming Architecture in System-Level CAD

This section introduces the implementation of the entire system architecture in a system-level commercial CAD with the goal of creating a digital model of the system and thus assessing the impact of non-idealities on the overall characteristics of the transmitter and receiver chains.

Each chain is composed of behavioral models representing the individual subsystem components, namely, the low-noise amplifier (LNA), the power amplifier, the local oscillator (LO), the mixer, and the analog-to-digital/digital-to-analog converters (ADCs/DACs). Additionally, the system includes both analog and digital filters at various stages of the architecture. All channels within the digital beamforming (DBF) network are identical in structure and their behavior is dictated by the known parameter spread.

The block diagrams of the transmitter and receiver systems, as implemented in the CAD system simulator, are reported in Figure 8a, the transmitter, and Figure 8b, the receiver.

The subsystems are modeled by built-in models taking into account the most relevant non-idealities of each element. These include I/Q mixer imbalances, amplifier compression, LO phase noise, effective number of bits of the ADC (analog-to-digital converter), as well as cross-coupling between antenna array elements. Concerning the latter, its effect is considered by including the cross-coupling matrix in terms of transmission scattering coefficients for each element. The matrix is then inserted between the block defining the array lattice and the receiver/transmitter front-end.

System-level performance is assessed through a comprehensive evaluation involving the radiation pattern under both single- and multibeam scanning configurations, the EVM, and the bit error rate (BER). These metrics respectively characterize the quality of the transmitter and receiver subsystems. Concerning EVM analysis, a virtual probe was integrated into the model to collect the signals feeding each array element. This setup enables the examination of how the individual contributions of the transmit chain combine to form the overall radiated waveform. By injecting a modulated test signal, the BER at the receiver back-end can be evaluated, thereby capturing the effects of receiver non-idealities arising from the superposition of signals across all channels. The radiation pattern is computed by mapping the element patterns across the antenna lattice and applying the corresponding excitation signals derived from the full chain simulation.

A summary of all simulation parameters is provided in Table 5. Hereinafter, these parameters are referred to as the nominal conditions.

### 3.1. Radiation Patterns Analysis

The first step is focused on evaluating the radiation pattern of the antenna to assess the influence of non-ideal feeding amplitudes and phases provided by the transmitting and receiving chains and the inter-element mutual couplings. The antenna lattice considered is the one reported in Figure 2b. All radiating elements share the same pattern, computed by full-wave e.m. analysis. Slight pattern modifications, which do occur from element to element in the array, are not considered since their effect is negligible. To assess the effect of inter-element coupling, we compare the antenna analyses carried out by considering or neglecting said couplings. Figure 9 presents the broadside radiation pattern results. It is evident that mutual coupling reduces the maximum gain by approximately 1.5 dB and causes a slight increase in side-lobe levels, particularly around $\pm 37^{\circ }$. The figure also includes a scenario with severely degraded performance due to an artificially intensified coupling effect. This extreme case is created by increasing by 50% the inter-element coupling from its nominal value, numerically computed. This is intended to represent a highly unlikely degraded case condition. Despite this increase in coupling, the radiation pattern metrics exhibit only about a 1 dB reduction in peak gain and a 1.5 dB increase in side-lobe level, which remains a satisfactory outcome under such severe impairment.

Table 6 provides a comprehensive numerical comparison of the analyses discussed above. The results indicate that, by increasing the beam steering angle, the SLL tends to rise, and this effect is stronger if mutual coupling effects are taken into account. Conversely, the array gain shows relatively less degradation under these conditions. These findings highlight the significant impact of element coupling on radiation pattern characteristics, particularly in off-broadside directions, underscoring the importance of considering coupling in accurate system-level performance evaluations.

Next, we report the results of the simulations in multibeam operation. For this purpose, 13 simultaneous beams are considered, with pointing directions as in Table 7. Beams are chosen at two values for ϕ angle to allow just two cuts to represent all of them. The choice does not impair generality.

Figure 10 shows the radiation pattern analysis in the multibeam case, both as a full gain map and two ϕ cuts. In the latter it is also possible to appreciate the scan loss, while the slight asymmetry, more apparent on the $45^{\circ }$ cut, is due to the corresponding asymmetry in the radiation pattern of the radiating elements, which, in turn, is due to the asymmetry in the feed pins (Figure 5d).

### 3.2. TX Chain Analysis

The results of the TX signal processing chain analysis are provided in terms of signal quality, i.e., its EVM. For this analysis, the chain was implemented considering the digital signal processor, the DAC, the mixer, the LO, and the PA; all of them defined by the parameters defined in Table 5.

The antenna (feed tapering and coupling for each array element) is included in the TX signal performance evaluations.

The test signal adopted for the transmitter chain testing consists of a 16-QAM (quadrature amplitude modulation) modulated signal, 12.5 MHz symbol rate, over-sampling factor equal to 8, and power level capable of driving at maximum power the PAs stage. The transmitter was evaluated both in the nominal state and in an “ideal” state. In the latter the subsystems behave ideally on the entire frequency range and signal dynamic. In ideal conditions the ENOB is set to 16, there are no imbalances, no phase noise, and mixer and LO are ideal. Only the PA linearity exhibits non-ideal, nominal, conditions. Table 8 reports the comparison between the two cases. It is apparent that, indeed, standard conditions are very close to the ideal case; thus, they can be both considered as a benchmark for further analyses, see Table 9.

Starting from the nominal state, the impact of the phase noise on the transmitted signal quality is analyzed first. Figure 11a, reports the influence of the phase noise on the EVM. Results are reported as a function of the phase noise amplitude, spanning the range −158 to −84 dBc/Hz at 1 MHz. From the figure it can be observed that, as expected at the nominal value of −120 dbc/Hz, the signal quality is consistent with the ideal case, while degrading the phase noise to −108 dBc/Hz and higher has a significant impact on the EVM. Concerning the impact of the DAC-induced noise and signal quality degradation, we considered the effective number of bits for the conversion. The simulation results for the transmitted signal are provided in Figure 11b, where a significant influence of the ENOB on the EVM is observed for values below 6 bit. Conversely, considering ENOB exceeding the nominal value, the EVM improves.

The next analysis considers the impact of the up-converter stage on the signal quality. For this purpose Figure 12 reports the influence of the up-converter imbalance in terms of amplitude (AM) and phase (PH) on the EVM. The EVM shows a stronger dependency on the magnitude imbalance rather than on the phase imbalance, and, for phase variations in excess of ±15% with respect to nominal values, (Table 5), its dependency on phase becomes negligible with respect to those related to amplitude.

The dynamic range of the transmit chain was analyzed to evaluate the impact of PA compression on the EVM.

The results of the analysis, maintaining the PA compression at the nominal state, namely the output 1 dB compression point of 27 dBm, and increasing the total channel power, are reported in Figure 13. The effect of increasing the transmitted signal power is evident in terms of a significant increase of EVM versus the PA output power level, starting from an output power of about 14 dBm. It is worth noticing that this level corresponds to about 38 dBm of total power delivered from the system to the entire 319-element array (not including the antenna gain). In order to evaluate the overall performance degrading with respect to subsystems degradation, we analyzed the EVM for subsystem parameters worse than the nominal ones. This could occur for different causes, for instance aging and temperature stress. For this purpose, we considered a PN of −108 dBc/Hz @ 1 MHz and ENOB = 4; in this case the EVM versus transmitted power is 0.1% worse than in nominal conditions until the PA power is below the compression point. Once the PA enters the compression region, the EVM is dominated by the PA nonlinearity itself and no variations are observed between the two cases; cf. Figure 13.

As an example of constellation degradation, Figure 14 shows the effect of compression for 22 dBm output power (Figure 14a), as well as for 30 dBm output power (Figure 14b), in nominal conditions.

### 3.3. RX Signal Chain

For the receiving chain, system performance was evaluated in terms of the bit error rate (BER) as a function of several critical parameters. The corresponding block diagram is shown in Figure 8b. The main subsystems of the RX chain include the antenna array (accounting for mutual coupling effects), the LNA, a down-conversion stage with a local oscillator, the ADC, the DBN, and a demodulator configured for BER estimation. The nominal values listed in Table 5 also apply to the receiver configuration.

The receiving chain assumes that the incoming signal arrives from the direction pointed by the antenna beam. The incident power on the array was simulated by considering an Incident Power Flux Density (IPFD) in the range [−63, −102] dBW/m^2^ corresponding approximately to [−66.5, −105.5] dBm available to each single element.

The first analysis considers the effect of inter-element coupling, which introduces an alteration to the waveform that modifies the BER to an unacceptable BER = 0.077.

The comparison of the received band-pass signal in both the cases with and without coupling is shown in Figure 15, reporting both magnitude, Figure 15a, and phase, Figure 15b. By relating the two time domain magnitudes and phase pairs, it is possible to find a complex coefficient for the received signal to compensate for the mismatches between the two couples of plots, which is due to antenna element coupling. This coefficient comprises both a magnitude scaling and a phase delay; it can be effectively computed by using a calibration process, which takes into account a simple pilot signal like, for instance, a BPSK signal. For the considered case—i.e., broadside—this coefficient is $1.0667\exp\left(j18^{\circ }\right)$. After this correction, the re-evaluated BER is equal to 0. The presence of this calibration signal is common in many communication systems, and follows policy and constraints depending on the radio access technology.

Figure 16 reports the influence of phase noise on the BER, considering the same phase noise values considered in the transmission chain. When the amplitude of the phase noise is higher than −110 dBc/Hz at 1 MHz offset from the carrier, the impact of phase noise on the BER is not negligible.

Figure 17 reports the influence of the LNA’s noise figure (NF) on the BER. Considering an NF equal to the nominal value—i.e., 1.4 dB—the BER is acceptable. Conversely, an NF equal to or exceeding 2 dB leads to unacceptable BER values.

In Figure 18, the impact of the effective number of bits (ENOB) on the BER is presented. The results indicate that when ENOB decreases below 4, the BER degrades significantly. Conversely, with an ENOB of 4 or higher, the BER remains within acceptable limits. These findings are particularly relevant in the context of system complexity reduction, as the power consumption of an ADC is a nonlinear function of both the sampling rate and ENOB. Therefore, minimizing the ENOB is essential for the efficient implementation of a digital beamforming receiver architecture.

The analysis presented above assumes a received input power of −66 dBm, the upper bound of the considered operating range. To extend this evaluation, an additional analysis is conducted for input power levels spanning the range [−66, −105] dBm. The primary objective of this investigation is to ensure adequate signal amplification prior to the ADC stage, thereby fully utilizing its dynamic range. To achieve this, the receiver chain incorporates a base-band variable gain amplifier (VGA).

An analysis of the BER as a function of the input power is performed with the subsystem parameters at their nominal value in order to evaluate the sensitivity. Results in Figure 19, blue curve, show that when the input level is below −95 dBm, the BER degrades and is not acceptable. An analysis of the BER was also carried out with the parameters degraded. Also in this case, we considered a PN of −108 dBc/Hz @ 1MHz and ENOB = 4. The results are reported for better comparison in Figure 19, red curve. Also in this case, the degraded subsystem features are representative of several factors, like, for instance, aging and operative large temperature variations.

## 4. Conclusions

The study has extensively analyzed a digital beamforming transceiver front-end for LEO satellite communications. It adopts the paradigm of the digital twin to assess how impairments affect the performance of the transmitting and receiving chains. For system antenna designers, such an evaluation highlights the most critical subsystem parameters of the RF and digital chain, thus permitting a trade-off between them. In the first part of the paper, we developed a thinned array antenna, implementing a specific tapering feed. The antenna was fully characterized. The radiation characteristics, along with the information about elements’ mutual coupling, were included in the system-level analysis of both the receiving and transmitting systems.

The system-level analysis highlighted the effect of the mutual coupling between antennas, and its twofold drawback: first, for the pattern, an overall gain reduction and a slight modification of the side-lobe level; second, an amplitude and phase scaling of the complex envelope that needs to be fixed at the receiver in the digital domain.

In terms of the transmitter front-end, the system-level analysis highlighted a set of values leading to an EVM better than 1.5% RMS considering standard conditions. In terms of the receiver front-end, the analysis highlighted a set of minimum features in terms of phase noise, ENOB, and I/Q imbalance, which provide a BER better than $10^{-4}$. The sensitivity can hardly be reduced below −95 dBm in the standard front-end setting; better sensitivity can be achieved with digital modulation coding.

An additional consideration is related to the impact of the antenna feed tapering on the average transmitting chain Power-Added Efficiency (PAE). In particular, according to the data reported in Table 3, the transmitting channels experience different peak power levels due to the feed tapering established during the antenna design. For this reason, using the same PA module for all the TX channels would result in a large number of those PAs operating with large power back-off, about 4.8 dB in our case; consequently, this results in a low average PAE. Scaling the PA size, the peak PAE would occur at the actual value of the peak power. According to the tapering strategy, we can consider that a significant number (N) of PAs are effectively driven with a reduced peak power. It results in those PAs reaching a maximum efficiency equal to (PAE-Δ). Thus, considering the total number of the channels ($N_{E}$), considering two sizes of PAs, the gain in average PAE can be calculated by(1)$${\mathrm{PAE}}_{Gain}=\frac{\mathrm{PAE}}{\left(\mathrm{PAE}-\Delta \right)+\frac{N}{N_{E}}\Delta }$$
where *N* out of $N_{E}$, is the number of PAs with a reduced size. The Figure 2 reports the normalized amplitude of the feeding with respect to the position of the array element in the lattice. Thus, the strategy of the PA down-scaling could consider that lower-sized PAs correspond to normalized feeding less than one. In the case under consideration, we can estimate PAE = 20%, Δ = 10%, *N* = 156 (i.e., the number of elements with a tapering coefficient less than or equal to 1), and $N_{E}$, being the PAE_*Gain*_, is about 1.33.

## Figures and Tables

**Figure 1 sensors-25-04645-f001:**
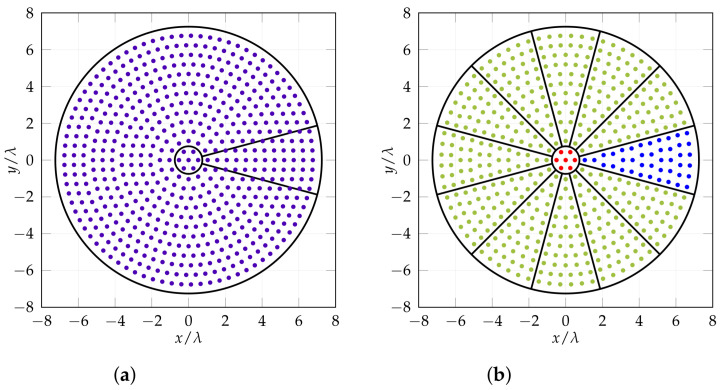
The full regular circular array composed of 13 rings and $N_{E}=566$ elements (**a**); and the sectored array of $N_{E}=511$ elements. (**a**) Circular regular array; (**b**) circular sectored array; red shows central sector $S_{0}$, blue shows $S_{I}$ sector, green ones are duplicate sectors.

**Figure 2 sensors-25-04645-f002:**
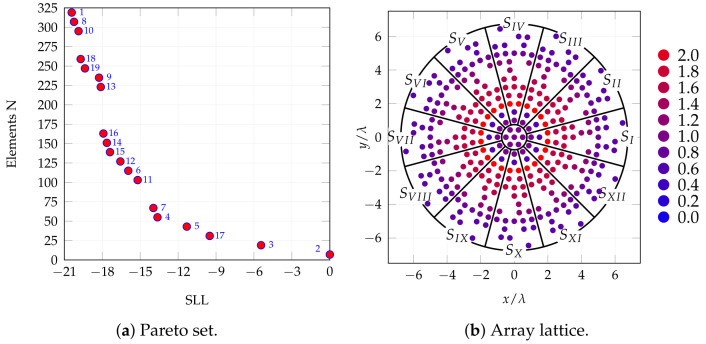
Optimization results. (**a**) A Pareto set derived from one of the many optimization runs with highlighted configuration 1, chosen for subsequent SO optimization; (**b**) geometry, with color-highlighted tapering (red: higher amplitude, blue: lower amplitude).

**Figure 3 sensors-25-04645-f003:**
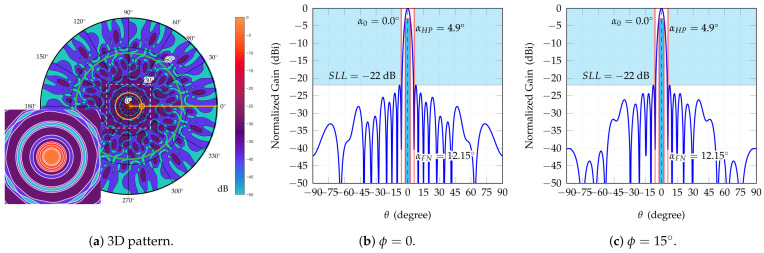
Attained broadside pattern. (**a**) Pattern drawn in polar coordinates; yellow crosshair shows the location of maximum-level side lobe; a small inset with a zoomed area shows a close-up of the broadside region $\left(\theta =0^{\circ },\phi =0^{\circ }\right)$ of the radiation pattern. (**b**) $\phi =0^{\circ }$ plane, (**c**) $\phi =15^{\circ }$ plane. Due to the symmetry of the array lattice, the cuts $\phi =30^{\circ }$, $\phi =60^{\circ }$, …$\phi =150^{\circ }$ are identical to the cut $\phi =0^{\circ }$, while the cuts $\phi =15^{\circ }$, $\phi =45^{\circ }$, …$\phi =165^{\circ }$ are identical to the cut $\phi =15^{\circ }$.

**Figure 4 sensors-25-04645-f004:**
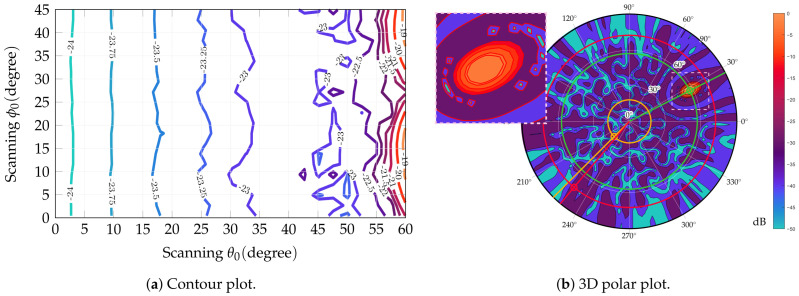
Capability of scanning of the proposed architecture. (**a**) Contour plot showing SLL in scanning for the optimized antenna; SLL computation is limited to a scan angle $57^{\circ }$ from broadside since eventual higher lobes out of this scan would fall outside Earth’s surface. (**b**) 3D polar plot for a scan angle $\theta_{0}=52.25^{\circ }$, $\phi_{0}=27^{\circ }$; yellow crosshair indicates highest SLL position within antenna footprint, red crosshair highest SLL outside footprint.

**Figure 5 sensors-25-04645-f005:**
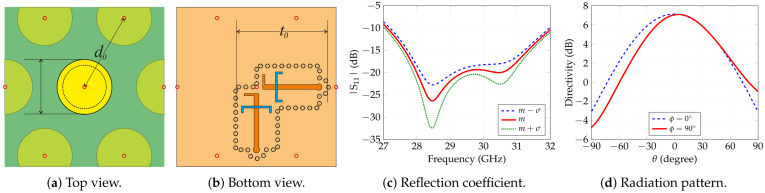
Radiating element design and performance. (**a**) Top view of the element showing the stacked structure. (**b**) Feeding network of the element. (**c**) Mean reflection-coefficient and standard deviation envelopes. (**d**) Gain pattern at center frequency (29.25 GHz).

**Figure 6 sensors-25-04645-f006:**
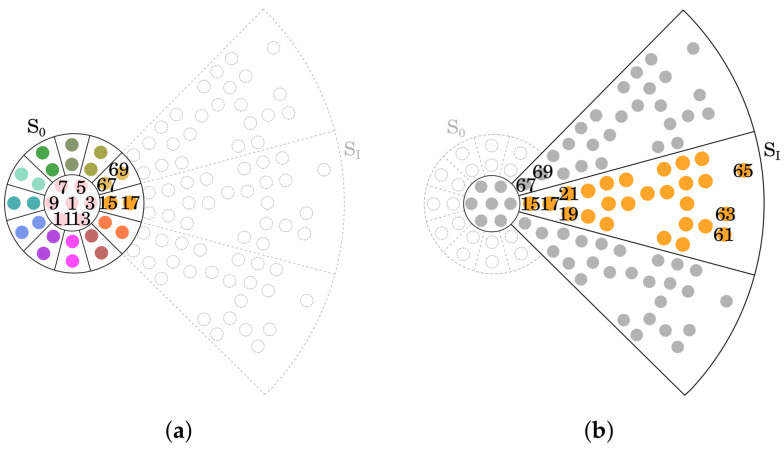
Elements considered for the mutual coupling analysis; not all numbers of feeding ports are reported in the figure. (**a**) $S_{0}$ Tile and immediate neighbors numeration. (**b**) $S_{I}$, but for symmetry, results apply to any outer tile, numeration.

**Figure 7 sensors-25-04645-f007:**
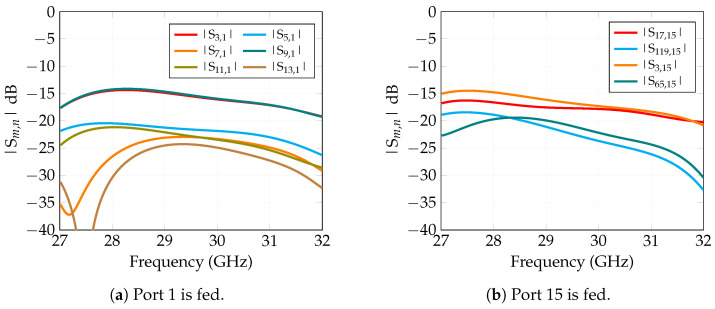
Behavior of the mutual coupling for both: the central sector when port 1 is fed (**a**) and the peripherical sector when port 15 is fed (**b**).

**Figure 8 sensors-25-04645-f008:**
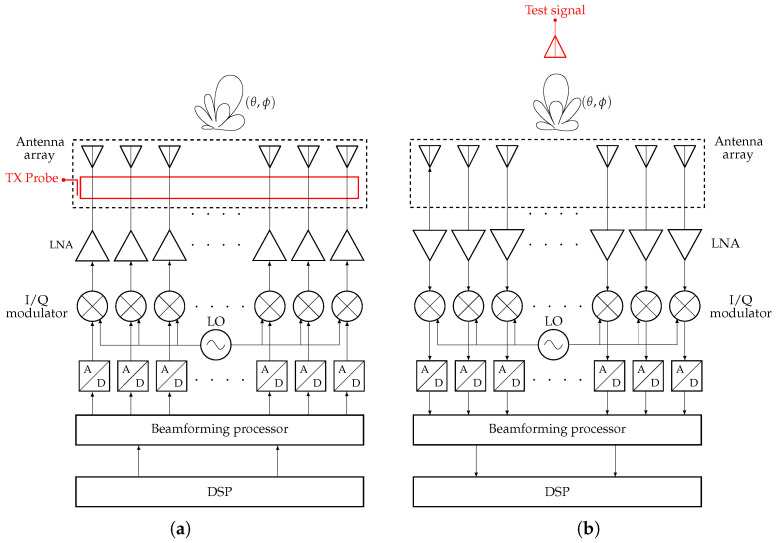
Block diagram of the digital beamforming system implemented in the CAD for the system-level analysis; suitable probe and test signal are highlighted in the picture. (**a**) Transmitter: a probe at the antenna array element section is highlighted. (**b**) Receiver: a test signal illuminating the antenna array is highlighted.

**Figure 9 sensors-25-04645-f009:**
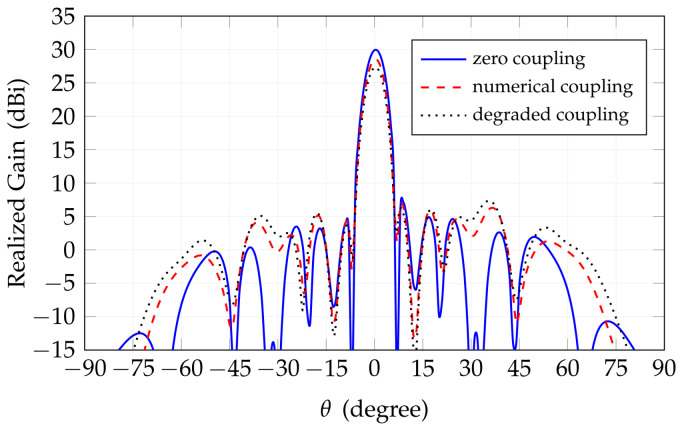
Effect of coupling between array elements in the array radiation.

**Figure 10 sensors-25-04645-f010:**
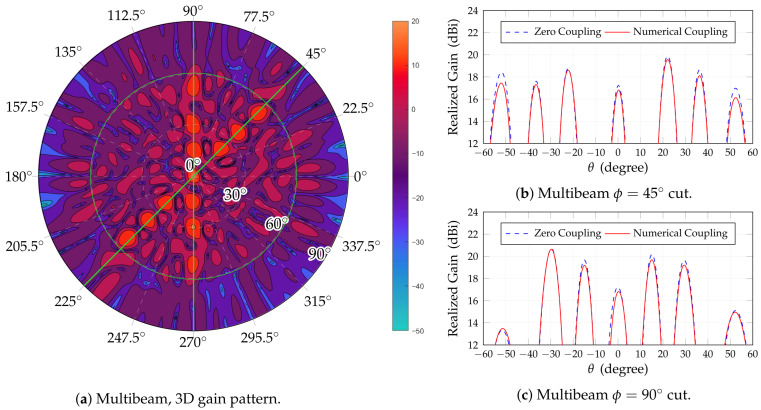
Multibeam performances for 13 beams. (**a**) Contour polar plot; (**b**) cut aligned along $\phi_{0}=45^{\circ }$; (**c**) cut aligned along $\phi_{0}=90^{\circ }$. Only the cases of no coupling and numerical coupling are considered for the cuts.

**Figure 11 sensors-25-04645-f011:**
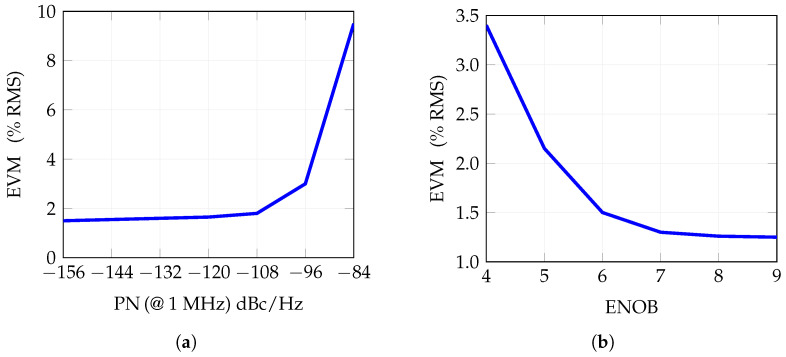
Influence of the EVM of phase noise and ENOB. (**a**) Influence of the phase noise—nominal condition considers phase noise = −120 dBc/Hz at 1 MHz offset from the carrier. (**b**) Influence of ENOB—nominal value is ENOB = 6.

**Figure 12 sensors-25-04645-f012:**
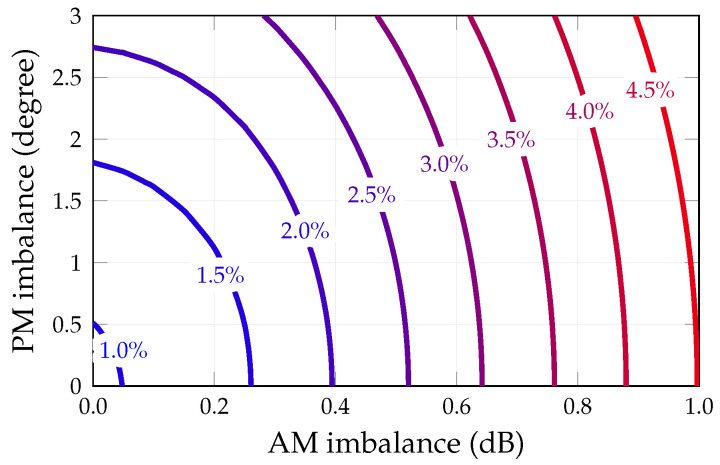
EVM, expressed as a percentage, obtained by varying both the imbalances of the upconverter.

**Figure 13 sensors-25-04645-f013:**
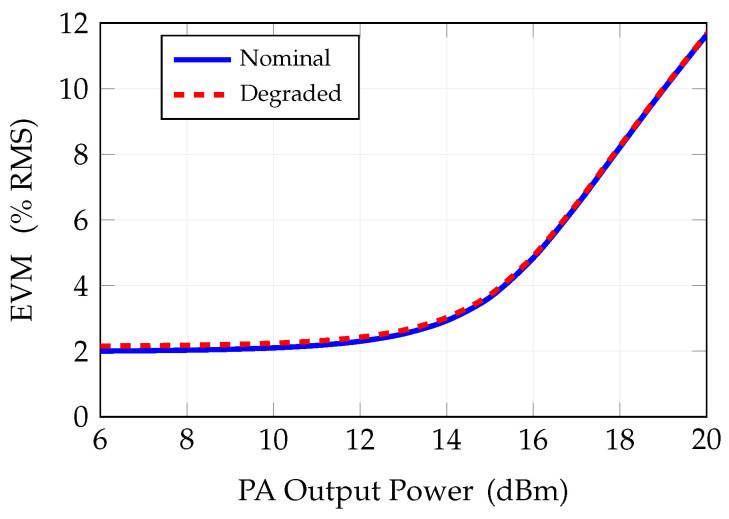
EVM versus output power at PA, in in nominal conditions and degraded.

**Figure 14 sensors-25-04645-f014:**
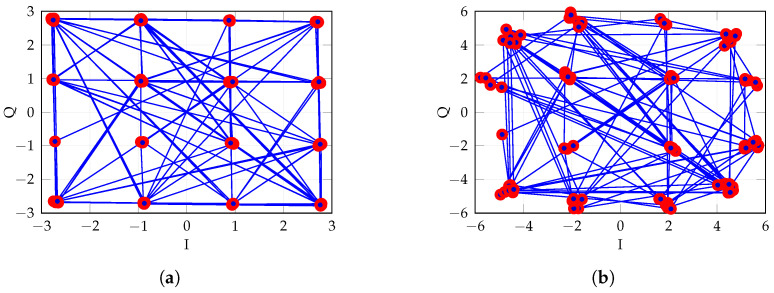
Constellations comparing different transmitting power levels. (**a**) Transmitting power 10 dBm (Nominal), EVM = 1.35%. (**b**) Transmitting power 18 dBm, EVM = 4.3%.

**Figure 15 sensors-25-04645-f015:**
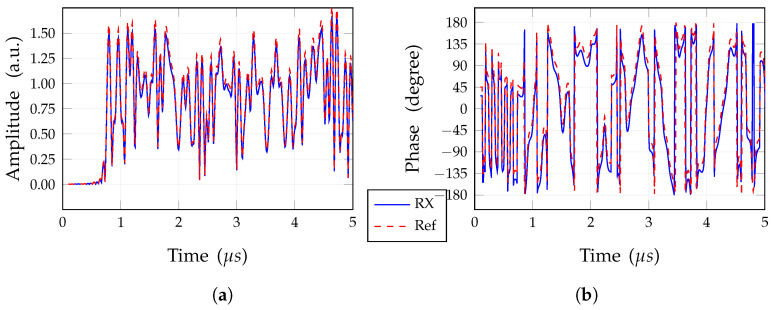
Waveforms of the ideal received signal (RX), with coupling, compared to the ideal received nominal signal (nominal). (**a**) Magnitude; (**b**) phase.

**Figure 16 sensors-25-04645-f016:**
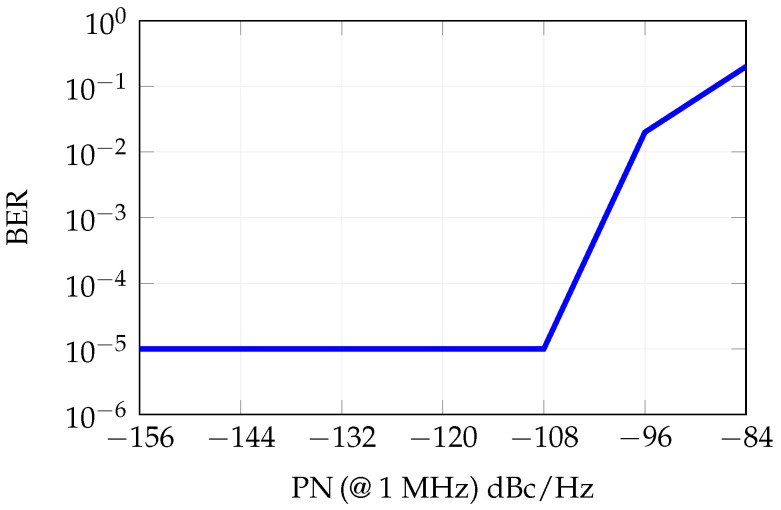
Influence of the phase noise on the BER, nominal conditions consider phase noise = −120 dBc/Hz at 1 MHz offset from the carrier.

**Figure 17 sensors-25-04645-f017:**
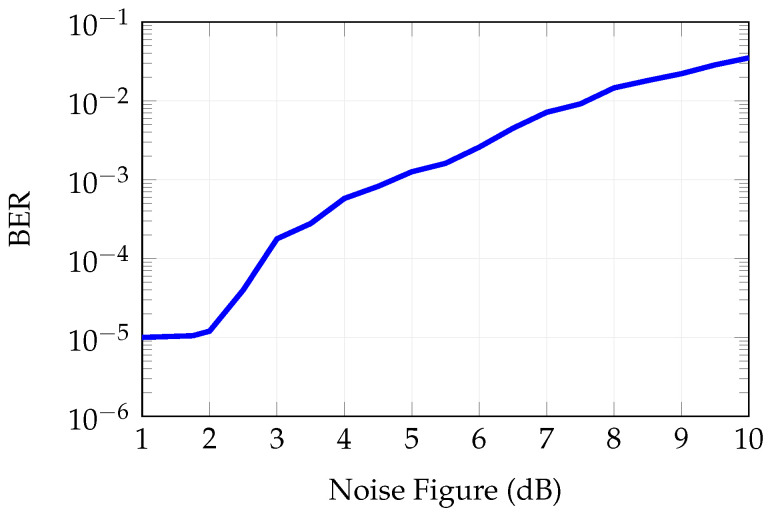
Bit error rate as a function of noise figure.

**Figure 18 sensors-25-04645-f018:**
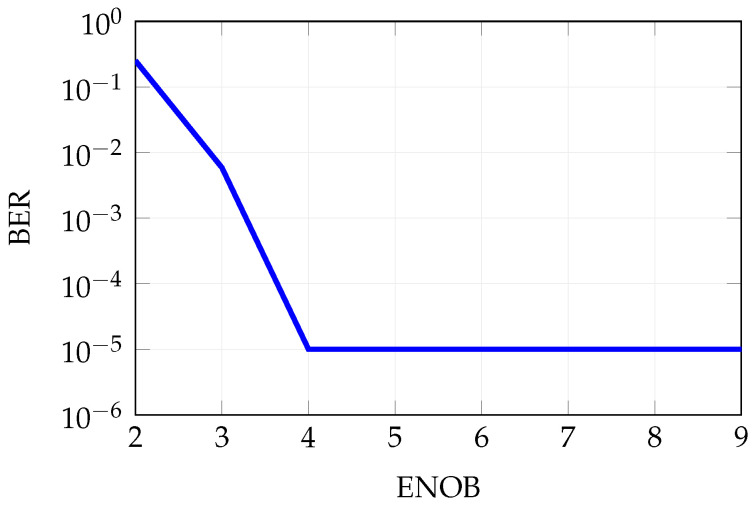
Influence of the ENOB on the BER, nominal condition is ENOB = 6.

**Figure 19 sensors-25-04645-f019:**
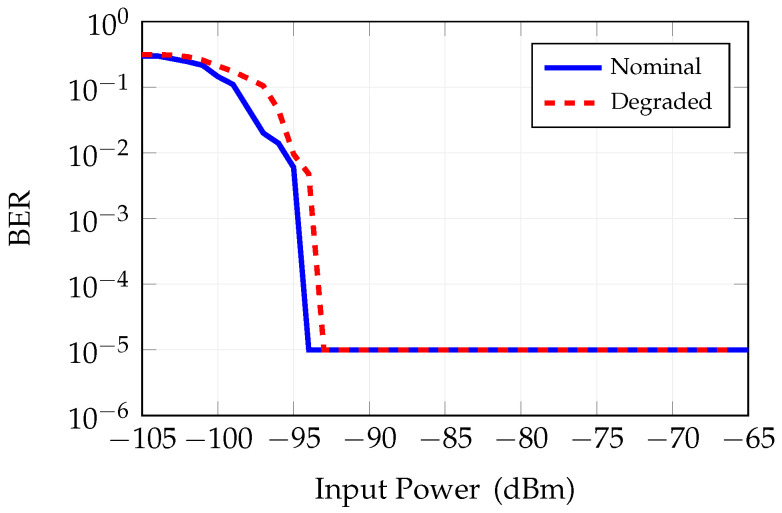
BER vs. input power in the RX chain (with nominal front-end parameter settings).

**Table 1 sensors-25-04645-t001:** Summary of actual and scheduled satellite services for IoT.

Operator	$N^{\circ }$ Satellites (Deployed)	Spectrum	Services	Operational
Starlink	12,000 LEO (6078)	Ku-band	Broadband	Yes
Kuiper	3236 LEO (2)	Ka-band	Broadband	Scheduled
Viasat	4 GEO (4)	Ka-band	Broadband	Yes
Hughenesat	3 GEO (2)	Ka-band	Broadband	Yes
Iridium	66 LEO (66)	L-band	messaging	Yes
One Web	648 LEO (648)	Ku-band	Broadband	Scheduled
Telesat	198 LEO (2)	Ka-band	Broadband	Scheduled

**Table 2 sensors-25-04645-t002:** Summary of requirements and constraints of the RX/TX antenna.

Requirements	Value
Operation frequency range	27.5–31 GHz
Coverage	$\pm 55^{\circ }$
SLL	<−25 dB
Grating lobes	Out of Earth
Directivity	>30

**Table 3 sensors-25-04645-t003:** Antenna optimization results. All cases are considered for 319 “on” elements. Bold line highlights the case selected for further analysis. SLL value is the worst SLL case computed while the main beam scans the whole range.

Concentric Rings	Tapered Elements	SLL (dB)	Dynamic $A_{\max}/A_{\min}$
5	168 (53%)	−20.85	2.55
8	240 (75%)	−20.94	2.12
10	288 (90%)	−21.61	4.21
**12**	**312 (98%)**	**−21.79**	**3.06**

**Table 4 sensors-25-04645-t004:** Amplitude tapering for the selected case of Table 3.

**Ring**	0	1	2	3	4	5	6	7	8	9	10	11	12	13
**Amp.**	1	1	0.93	0.64	1.96	1.45	1.52	1.41	1.15	1.10	0.83	0.70	0.73	0.69
	Fixed		Tapered											

**Table 5 sensors-25-04645-t005:** Nominal parameters adopted in simulations. Abbreviations: Param. = Parameter, Val. = Value, PN = Phase Noise, ENOB = Effective Number of Bits, I/Q AM and PH = I/Q Amplitude and Phase Imbalance. Units: Gains, noise figures, and imbalances are in dB; output powers (OP1dB, OIP3) are in dBm; PN values are in dBc/Hz. Frequency is in Hz (as specified).

PA		LO		ADC/DAC		IQ Mod.		LNA	
Param.	Val.	Param.	Val.	Param.	Val.	Param.	Val.	Param.	Val.
Gain	12	PN @ 10 kHz	−70	ENOB	6	I/Q PH	$1.7^{\circ }$	Gain	29
OP1dB	27	PN @ 100 kHz	−96			I/Q AM	0.07	NF	1.4
OIP3	40	PN @ 1 MHz	−120					OP1dB	6
								OIP3	11

**Table 6 sensors-25-04645-t006:** Antenna gain (G) and the side-lobe level (SLL) comparison.

Cases	$\theta_{0}=0^{\circ }$		$\theta_{0}=30^{\circ }$		$\theta_{0}=52^{\circ }$	
	G (dB)	SLL (dB)	G (dB)	SLL (dB)	G (dB)	SLL (dB)
zero coupling	29.98	22.48	29.68	24.68	27.69	21.57
numerical coupling	28.55	22.25	29.28	21.42	27.15	19.45
degraded coupling	27.32	21.11	28.83	18.87	26.45	19.52

**Table 7 sensors-25-04645-t007:** Directions used for multibeam analysis.

$\theta_{0}$	$\pm 52^{\circ }$	$\pm 30^{\circ }$	$\pm 15^{\circ }$	0	$\pm 52^{\circ }$	$\pm 37^{\circ }$	$\pm 22^{\circ }$
** $\phi_{0}$ **	$90^{\circ }$	$90^{\circ }$	$90^{\circ }$	N.A.	$45^{\circ }$	$45^{\circ }$	$45^{\circ }$

**Table 8 sensors-25-04645-t008:** Characteristics of the transmitted signal, versus nominal and ideal transmitting chain.

Parameter	Nominal State	Ideal State
EVM (% RMS)	1.54	1.54
EVM (peak)	3.36	2.54

**Table 9 sensors-25-04645-t009:** Antenna gain (G) and side-lobe level (SLL) for the full TX chain, considering all components with nominal parameters, compared with the previous results given in Table 6 (pertinent row is duplicated for easier comparison).

Cases	$\theta_{0}=0^{\circ }$		$\theta_{0}=30^{\circ }$		$\theta_{0}=52^{\circ }$	
	G (dB)	SLL (dB)	G (dB)	SLL (dB)	G (dB)	SLL (dB)
Nominal conditions	28.86	21.11	29.17	18.17	26.63	15.63
Numerical coupling	28.55	22.25	29.28	21.42	27.15	19.45

## Data Availability

Data are contained within the article. Additional data can be provided—if not restricted—by mailing with corresponding authors.

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
