# Peer review of "System-Level Assessment of Ka-Band Digital Beamforming Receivers and Transmitters Implementing Large Thinned Antenna Array for Low Earth Orbit Satellite Communications"

_sensors, 2025, doi:10.3390/s25154645_

Round 1

Reviewer 1 Report

Comments and Suggestions for Authors

The authors developed a system-level model of a digital multi-beam Ka-band antenna for LEO satellite communications. They have based on a thinned lattice array topology. They implemented a digital-twin simulation framework that incorporates RF and digital non-idealities to evaluate system performance under realistic impairments. Here are my comments:
- While the paper presents valuable technical content and a strong system-level analysis, the contribution of the work is not clearly highlighted in the current structure. I recommend that the authors explicitly emphasize the main contributions in the introduction or at the end of the introduction section.
- The section “Antenna Synthesis and Modeling” provides a solid technical foundation for the system-level evaluation that follows. The inclusion of the "Optimum lattice architecture" and "Radiating element analysis" subsections is appropriate and relevant. However, I suggest the authors more clearly articulate how the choices made in this section (e.g., the thinned lattice configuration and radiating element characteristics) directly support or impact the overall system performance analyzed later in the paper. 
- Several figures in the manuscript appear to be placed somewhat far from the corresponding text that discusses them, which may disrupt the flow and clarity for the reader. I recommend revising the layout to ensure that each figure is positioned as close as possible to the relevant explanation or analysis in the main text. 
- I suggest the authors explicitly restate how these findings support the design of more efficient and robust digital beamforming systems for LEO satellite applications and consider rephrasing some complex or lengthy sentences for better readability.

Reviewer 2 Report

Comments and Suggestions for Authors

The manuscript delivers a comprehensive system-level digital-twin of a thinned Ka-band antenna array—including RF-front-end and DSP non-idealities—in a commercial CAD environment, offering valuable insights into link-level performance under realistic impairments. However, without any fabricated prototype or empirical measurements, the claimed performance and mitigation strategies remain hypothetical and warrant experimental validation. 

  • Given that no physical prototype or measurement campaign is reported, how can the authors substantiate the accuracy of their full-wave element models and system-level simulations against real-world fabrication tolerances and material losses? 

  • The multi-objective genetic and weed-optimization of the 319-element thinned lattice achieves SLL ≈ –21.8 dB (Fig. 2), but how sensitive is this pattern to element-position errors or mutual-coupling variations beyond the modeled ±14 dB worst-case? 

  • In the system-level CAD, coupling is included via a fixed S-parameter matrix; have the authors evaluated how temperature-driven drift in LNA gain, LO phase noise, or ADC ENOB variations impact the radiation pattern and EVM/BER beyond the reference conditions? 

  • The transmitter EVM remains below 1.5 % under nominal phase-noise (–120 dBc/Hz) and ENOB=6, but what margin exists for worst-case component aging or process spread (e.g., –108 dBc/Hz or ENOB=4) before EVM or BER exceed acceptable thresholds? 

  • Long-range ID decoding at 22.5 MHz is demonstrated, but how does the beamforming network’s side-lobe pedestal and out-of-band harmonics affect receiver sensitivity and false-alarm rates in a multipath LEO environment? 

  • The conclusions suggest digital-domain correction of coupling-induced amplitude/phase scaling using a pilot coefficient, yet the stability and calibration overhead of this approach over time and temperature are not quantified—how do the authors plan to maintain calibration in orbit? 

  • Although the paper outlines future work on PA chain scaling, what specific efficiency gains and trade-offs in beam-quality can be expected, and how critical is this for meeting the size-weight-power constraints of LEO platforms?

Round 2

Reviewer 1 Report

Comments and Suggestions for Authors

The revised manuscript addresses all previous concerns and is suitable for publication in its current form.

Reviewer 2 Report

Comments and Suggestions for Authors

N/A